# Prevalence of Schistosomiasis and Soil-Transmitted Helminthiasis and Their Risk Factors: A Cross-Sectional Study in Itilima District, North-Western Tanzania

**DOI:** 10.3390/life13122333

**Published:** 2023-12-12

**Authors:** Jungim Lee, Seungman Cha, Yoonho Cho, Anold Musiba, Boniphace Marwa, Humphrey D. Mazigo

**Affiliations:** 1World Vision Korea, Yeouinaru ro, Youngdeunpo–gu, Seoul 07327, Republic of Korea; jungim_lee@worldvision.or.kr (J.L.); yoonho_cho@worldvision.or.kr (Y.C.); 2Department of Global Development and Entrepreneurship, Graduate School of Global Development and Entrepreneurship, Handong Global University, Pohang 37554, Republic of Korea; seungman.cha@handong.edu; 3District Medical Office, Itilima District Council, Itilima P.O. Box 308, Simiyu, Tanzania; anoldmusiba@yahoo.com; 4Simiyu Regional Medical Office, Bariadi 39101, Simiyu, Tanzania; marwaboniphace3@gmail.com; 5School of Public Health, Catholic University of Health and Allied Sciences-Bugando, Nyamagana P.O. Box 1464, Mwanza, Tanzania

**Keywords:** schistosomiasis haematobium, schistosomiasis mansoni, prevalence, soil-transmitted helminthiasis, Tanzania

## Abstract

Schistosomiasis and soil-transmitted helminthiasis remain a public health concern in Tanzania. This study investigated the prevalence and intensities of *Schistosoma haematobium*, *S. mansoni*, and soil-transmitted helminths and associated factors in Itilima district, north-western Tanzania. A cross-sectional survey was conducted between August and September 2020 among 3779 primary schoolchildren in 62 primary schools and 1122 adults in 19 villages. Urine samples were obtained from each participant and examined visually for the presence of macrohaematuria, microhaematuria, and *S. haematobium* eggs using a urine dipstick and urine filtration test. A single stool sample was obtained from each participant and screened for *S. mansoni* and soil-transmitted helminths using the Kato Katz and formalin-ether concentration techniques. A questionnaire was administered to schoolchildren to elucidate the risk factors for schistosomiasis. The overall prevalence of *S. haematobium* in adults was 8.1% (95% confidence interval (CI), 6.6–9.8%). In total, 3779 schoolchildren had complete results from urine testing, and the overall prevalence of S. *haematobium* was 10.1% (95% CI, 9.1–11.1%). The prevalence of S. *mansoni* and soil-transmitted helminths was relatively low among both children and adults compared to S. *haematobium*. Factors associated with *S. haematobium* infection among schoolchildren were the mother’s occupation, children aged 11–15 years, and water contact behaviour. The odds of having schistosomiasis infection among children aged 11–15 are 40% higher than those aged 5–10 (95% confidence interval (CI), 10–80%, *p* = 0.04). Children of parents who are livestock keepers have 12.3 times higher odds of having infection compared to those who have small-scale businesses (95% CI, 1.0–5.4, *p* = 0.03). Children who are in contact with infested water more than three times a week have 2.1 times higher odds of having an infection compared to those who do not (95% CI, 2.1; 1.6–2.8, *p* < 0.001). The findings provide updated geographical information on prevalence, yielding insights into the planning and implementation of mass drug administration in rural Tanzania.

## 1. Introduction

Soil-transmitted helminth (STH) infections are among the most prevalent infections globally, predominantly affecting the poorest and most marginalized communities. These infections are transmitted through eggs found in human faeces, which subsequently contaminate the soil in regions with inadequate sanitation [1]. The primary species that infect humans include the roundworm (*Ascaris lumbricoides*), the whipworm (*Trichuris trichiura*), and hookworms (*Necator americanus* and *Ancylostoma duodenale*) [1,2]. Over 1.5 billion people, equating to 24% of the global population, are infected with STHs [1]. These infections are extensively distributed in tropical and subtropical regions, with the highest incidences reported in Sub-Saharan Africa, the Americas, China, and East Asia [1].

In endemic areas, pre-school-age children and school-age children exhibit the highest prevalence and intensity of STH infections. Current data suggest that over 270 million pre-school-age children and more than 600 million school-age children globally reside in areas with intense transmission of these parasites [1,2,3]. Among this group, the infections are linked to symptoms such as diarrhoea, vomiting, anaemia, and dehydration, which can result in growth retardation and diminished cognitive function [1,3]. These negative health impacts contribute to reduced educational performance in childhood and increased school absenteeism [1].

Schistosomiasis continues to be one of the most widespread neglected tropical diseases (NTDs), resulting in significant morbidity in Sub-Saharan Africa (SSA) [4]. The disease in this region is commonly caused by the *Schistosoma haematobium* and *S. mansoni* species. Globally, it is estimated that schistosomiasis affects over 290 million people [4], with more than 779 million residing in areas of high transmission [4]. Among those infected, 93% are in SSA [4], and about 76% live in high-transmission zones. It is estimated that 120 million people exhibit symptoms related to schistosomiasis, and the disease is responsible for over 2.8 million years lived with disabilities [4]. School-age children have the highest prevalence and intensity of schistosome infections, while adults typically exhibit a lower infection intensity [4]. Recent reports indicate that pre-school-age children from SSA are also infected and exhibit a high intensity of infection [5].

The strategy for controlling STH and schistosomiasis infections hinges on mass drug administration (MDA) of preventive medication, primarily albendazole (ALB), mebendazole (MEB), and praziquantel (PZQ), to children at risk. This approach was initiated over a decade ago [6,7]. The World Health Organization (WHO) recommends annual treatment if the baseline prevalence of STH in the community exceeds 20% but is less than 50% and for schistosomiasis when the prevalence exceeds 10% [6,7]; the main goal is to reach at least 75% of the targeted population. If the prevalence of STH and schistosomiasis infections in the community surpasses 50%, treatment should be administered twice a year [6,7]. The WHO has established a global objective to eradicate morbidity due to STH and schistosomiasis infections in children by 2023 [1,2]. This goal is to be achieved by consistently treating at least 75% of the children in endemic areas [6,7,8].

The WHO, through resolutions WHA 65.21 and 66.12 in 2012 and 2014, encourages national control programmes to shift their focus from morbidity control to the broader goals of eliminating schistosomiasis and STH in endemic countries. The ultimate objective is to interrupt the transmission of schistosomiasis [7,8]. To achieve this, it is necessary to identify areas with persistently high infection rates, characterize these areas, and implement treatment strategies that go beyond what has been achievable solely through multiple rounds of MDA [9]. Furthermore, to monitor the effectiveness of the MDA approach, it is crucial to understand the baseline prevalence of the disease. This understanding will enable the measurement and evaluation of the impact of interventions. However, this is only possible if the nature of transmission and prevalence of the disease at the local level are known, particularly in the post-MDA period [10,11,12]. In response to this need, the WHO has emphasized the importance of creating predictive and precision maps to anticipate the distribution of schistosomiasis and STHs. The geographical information system has been embraced in disease control programs to map the spatial patterns of human diseases, thereby improving the efficient allocation of available transmission control interventions [13].

The overall aim of this study was to determine the geographical prevalence and infection intensity of schistosomiasis and STHs in the Itilima district of north-western Tanzania and to create precise prevalence maps for these diseases. Our goal with this research was to establish a solid foundation for the planning of comprehensive mass preventive chemotherapy and integrated interventions. These were to be implemented within the framework of the National Neglected Tropical Diseases Control Programme, with the ultimate aim of achieving the elimination of schistosomiasis and STHs in communities where these diseases are highly endemic.

## 2. Materials and Methods

### 2.1. Study Area

This study was conducted in the Itilima district of the Simiyu region in north-western Tanzania, situated at longitude −3.73333 and latitude 33.48333. The district is bordered by the Bariadi district to the north, the Ngorongoro district to the east, the Maswa and Meatu districts to the south, and the Magu district to the west. As per the 2012 population census, the district had a population of 313,900 people. Data from the National Neglected Tropical Disease Programme suggest that the district is highly endemic for *S. haematobium* infection [14]. The district exhibits varying levels of *S. haematobium* endemicity (prevalence) and continues to experience high transmission rates in certain villages, despite repeated rounds of MDA [12,15]. The district’s environmental conditions are conducive to the transmission of *S. haematobium* [15]. According to data from the National Strategic Plan for Controlling Schistosomiasis, the overall prevalence of *S. haematobium* infection exceeds 50% in this district [14,16]. The district experiences annual rainfall ranging between 930 and 1200 mm, and temperatures typically fall within the range of 25–28 °C. The highest temperatures are usually recorded between August and October. The primary economic activities of the district’s inhabitants include farming, livestock keeping, and small-scale business. Data on STHs and *S. mansoni* are currently unavailable, and this study aims to address this gap.

### 2.2. Study Design, Inclusion, and Exclusion Criteria

A school and community-based cross-sectional design was used for this district-wide survey, based on the recommendation of the World Health Organization [17]. This study involved schoolchildren aged 3–17 years who were attending selected schools within the district, as well as the adult population (aged 15–60 years) residing in the village where these schools are located. The inclusion and exclusion criteria used for participant recruitment are detailed in the Table 1 below.

### 2.3. Sample Size and Sampling Method

The sample size and sampling strategies were adapted from the study conducted by Cha et al., 2017 [17]. A probability-proportional-to-size approach was utilized for the selection of schools. Upon selecting the schools, 60 students (30 boys and 30 girls) were sampled from each institution [17]. The WHO recommends the selection of 50 schoolchildren from each school to be included in the study [17]. Considering a 16% non-response rate, an additional 10 students were included beyond the WHO’s recommendation. A systematic random sampling technique was employed to select pre-school- and school-aged children to participate in the study. The selection ranged from pre-school classes up to class six (6). This technique has been detailed elsewhere [13].

A two-stage random sampling method was used at the district level to select participating schools/communities and children for the study. This approach aimed to generate accurate prevalence estimates [17]. Initially, the district was divided into one to three distinct ecological zones based on their proximity to bodies of water (near, less than 1 km; medium, 1–5 km; and far, more than 5 km). Ecological zones are characterized as areas situated at similar distances from water bodies within a given locality [17]. Subsequently, schools/communities were sampled from each ecological zone. Finally, schoolchildren and adult community members were sampled from the chosen schools and communities.

### 2.4. Data Collection

#### 2.4.1. Recruitment, Enrolment, and Retention of the Study Participants

The recruitment and sampling for our study were conducted within school environments for children and agreed-upon community areas for adults. Our study team visited the district in question to plan and hold informative meetings with the district and ward/village/community authorities or leaders. We also conducted community and school meetings to raise project awareness among community members and parents/guardians. The school and village administrations were informed about the project and the scheduled dates for sampling visits. In our epidemiological studies, we collaborated closely with Environmental Health Officers, head teachers, matrons, patrons, chairpersons of the village health committee, school board members, and village/ward executive officers. All of these individuals reside within the community, making them crucial contact persons who assist in disseminating information. The recruitment of study participants took place in the school environments for children and at the agreed-upon community site for adults. Parents/guardians of schoolchildren were invited to an informative meeting at the school two days prior to the sampling. This meeting was held to explain the study procedures, treatment, and the importance of obtaining written informed assent from the children and informed consent from the adults for participation in the study.

#### 2.4.2. Resources, Facilities, and Staff Available to the Research

The fieldwork was conducted in collaboration with the Catholic University of Health and Allied Sciences in Mwanza, Tanzania, and World Vision Tanzania. Urine and stool samples were collected and immediately processed at the field site’s laboratory, which was established in one of the school buildings, and examined using a light microscope. After the preparation of Kato Katz thick slides, the remaining stool samples were preserved using 10% formalin. These samples were further examined for the presence of STHs and *S. mansoni* eggs using the formalin-ether concentration technique at the CUHAS laboratory following the fieldwork.

### 2.5. Data Collection Strategies

#### 2.5.1. Questionnaire

Face-to-face interviews were conducted with each selected child and adult using a pretested questionnaire. This questionnaire gathered demographic information about the study participants. The children were questioned about general sanitation and environmental conditions, as well as their behaviours related to water contact or open defecation (Appendix A, Questionnaire set). For adults, only their sex and age were recorded, and no additional questions were administered.

#### 2.5.2. Parasitological Examination of Stool and Urine Samples for Schistosomiasis Infections

a.Parasitological examination of *S. mansoni* eggs using the Kato Katz technique

A single stool sample was collected from each child participant at baseline, using a sealed, labelled stool container. These collected stool samples were then processed using Kato Katz thick smears, with a template of 41.7 mg per thick smear. Four Kato Katz thick smears were prepared from each stool sample and examined by two independent laboratory technicians who were experienced in the Kato Katz technique. Additionally, after a 24 h period, the Kato Katz smears were examined for the presence of other STH and *S. mansoni* infections. To ensure quality, 20% of all positive and negative Kato Katz thick smears were re-examined by a third laboratory technician, who was blinded to the results of the initial two technicians.

b.Parasitological examination of *S. haematobium* infection using urine filtration techniques

A single urine sample was collected from each school-aged child and adult participating in the study. The collection took place between 10:00 a.m. and 2:00 p.m. in the agreed-upon school environment. Each collected urine sample was thoroughly examined for macro-haematuria, and a urine dipstick/urinalysis reagent strip (Mission^®^, Expert, San Diego, CA, USA) was used to determine the presence of micro-haematuria. A filtration technique was employed for the screening of urine samples, and light microscopy was utilized to examine the urine filters for the presence of *S. haematobium* eggs [18].

c.Parasitological screening of STHs using the formalin-ether concentration technique

The formalin-ether concentration technique (FECT) is a widely utilized sedimentation method for diagnosing intestinal protozoa in preserved stool samples [19]. The most frequently used fixatives for stool preservation are either formalin or sodium acetate-acetic acid-formalin (SAF) [19]. The stool samples left over after the preparation of Kato Katz smear processing were preserved using 10% formalin. The procedures for the formalin-ether concentration technique were conducted at the Catholic University of Health and Allied Sciences laboratories. In brief, approximately 1 g of the collected stool samples from each child was preserved in 10 mL of SAF solution, which contained 10% of formaldehyde solution. The samples were then roughly broken up, and large faecal particles were removed by straining through medical gauze or a tea sieve into a conical tube [19]. The collected filtrate was vigorously shaken with 3 mL of diethyl ether, followed by centrifugation at 2000–5000 rpm for 2–3 min. Next, three layers of the supernatant were discarded, and the sediment was transferred to a glass slide. A drop of povidone-iodine was added to stain the eggs, and a microscopic examination was performed [19]. If the final sediment contained a volume of more than 1 mL, the first two steps were repeated and part of the suspension was removed. Subsequently, 2–3 mL of diethyl ether was added to the remaining sediment. The tube was sealed with a rubber stopper, shaken vigorously for approximately 30 s, and then centrifuged at 2000–5000 rpm for 2–3 min. Eggs of the STH and *S. mansoni* were counted under a low-power objective and recorded in the laboratory book.

#### 2.5.3. Geographical Distribution of Infection

To determine the geographical distribution of infection prevalence, we mapped the locations of all participating villages and primary schools using GPS software installed on a mobile phone. We then imported all collected coordinates into ArcView GIS Software (GIS 3.3x)(ESRI- https://www.esri.com/en-us/home, accessed on 5 December 2023), which enabled us to generate maps of the district.

### 2.6. Data Analysis

Data were double-entered into a Microsoft Excel sheet, cleaned, and then exported to Stata version 15 (StataCorp, College Station, TX, USA). The primary objective of the analysis was to ascertain the prevalence of STHs, *S. haematobium*, and *S. mansoni*, utilizing the Kato Katz technique, urine filtration technique, and formalin-ether concentration technique. Continuous variables, such as age and egg intensities, were summarized using the mean ± standard deviation (SD). Frequencies, proportions, and categorical variables were compared using either the chi-square (χ^2^) or Fisher exact tests, while continuous variables were compared using the t-test. For *S. mansoni* eggs, the arithmetic means of egg counts were derived from the counts of two Kato Katz smears and then multiplied by 24 to determine the individuals’ eggs per gram of faeces. The intensity of infection was categorized according to WHO criteria, with 1–99 epg, 100–399 epg, and ≥400 defined as low, moderate, and heavy intensities of infection, respectively [9]. For *S. haematobium* infection, the geometric mean egg output was estimated based solely on infected participants. Infection intensities were classified into two categories as per WHO recommendation [20]: (i) light infection (<50 eggs/10 mL of urine) and (ii) heavy infection (≤50 eggs/10 mL of urine). Bivariate and multivariate logistic regression analyses were employed to assess factors associated with the helminth infections diagnosed in the study. In the bivariate analysis, factors with *p* ≤ 0.2 were considered for multivariate analysis. Adjusted odd ratios, along with their 95% confidence intervals, are presented.

### 2.7. Ethical Considerations

Ethical approval for this study was sought from the Joint Institutional Review Board of the National Ethical Committee and the Lake Zone Institutional Review Board. Permission for this study was also obtained from the Regional and District Administrative Authorities of the Simiyu region and Itilima district. Two days prior to their participation in the study, children were given informed consent forms, translated into Kiswahili, to take home to their parent(s) or guardian(s). Parents were then invited to the school on the day of screening to provide their consent. An assent form was also developed for children aged 9–17 years to read and understand the study’s procedures and objectives. All adults participating in the study were provided with informed consent forms prior to their participation. To ensure confidentiality, all clinical and demographic data from the study participants were securely stored in a locked cabinet, and all participants were identified using codes. Any children or adults identified as being infected with *S. haematobium*, STH, or *S. mansoni* by any of the diagnostic tests used in the study were treated with ALB and PZQ (40 mg/kg), which is in accordance with WHO recommendations [9].

## 3. Results

### 3.1. Schoolchildren’s Results

#### 3.1.1. Demographic Information

A total of 3801 schoolchildren from 62 primary schools in various wards of the Itilima district council participated in the current study. Of these children, 50% (1889/3779) were female. The mean age of the students was 11.2 ± 2.6 years. The general characteristics of the students are presented in Table 2.

#### 3.1.2. Prevalence and Infection Intensity of *S. haematobium* and Microhaematuria

A total of 3779 schoolchildren had complete results from the urine filtration technique, and the overall prevalence of *S. haematobium* was 10.1% (95% CI: 9.1–11.1%). Male children had a significantly higher prevalence than female children (11.3% versus 8.9%, χ^2^ = 6.71, *p* < 0.01). The oldest age group (16–17 years) had a higher prevalence than the younger age groups (χ^2^ = 12.11, *p* < 0.002). The overall prevalence of microhaematuria was 4.9% (190/3801), with no sex differences (χ^2^ = 2.5345, *p* = 0.11).

Figure 1 shows the prevalence of schistosomiasis by age group among schoolchildren in Itilima district, north-western Tanzania.

Based on the World Health Organization intensity categories, 79.7% (303/380) had low (1–50 eggs/10 mL) and 20.3% (77/380) had heavy (>50 eggs/10 mL) infection intensities.

#### 3.1.3. Prevalence and Infection Intensity of *S. mansoni*, STHs, and Other Parasitic Infections

A total of 3779 schoolchildren were tested using the Kato Katz method. The overall prevalence of *Schistosoma mansoni* was found to be 0.29% (95% CI, 0.15–0.52%), and the overall geometric mean egg (GMepg) intensity was 45.4 GMepg (95% CI, 21.5–95.7). The majority of children infected with *S. mansoni* exhibited light infection intensity (72.7%), and the others (27.3%) had moderate intensity. There was no case of heavy infection.

In the case of soil-transmitted helminth infections, eggs of hookworms were detected in three children, while one child was found to have eggs of *T. trichiura*. Additionally, another parasitic infection identified was *Entamoeba histolytica/Entamoeba dispar*, with a prevalence of 25.4%.

#### 3.1.4. Variations in the Prevalence of *S. haematobium* among Primary Schools Involved in the Study

The prevalence of *S. haematobium* varied significantly among primary schools, with rates ranging from 0% to 45.8%. (Figure 2) The schools with the highest prevalence, exceeding 30%, were Mitobo (31.7%), Budalabujiga A (32.8%), Budalabujiga B (33.3%), Lung’wa (36.1%), Kashishi (43.5%), Mwanunui (45.5%), and Zegeyu (45.8%). Other schools, including Gaswa (22.9%), Nkuyu (27%), and Gambasingu A (28.3%), had prevalence levels between 20% and 30%. A total of 31 schools had prevalence levels ranging from 1% to 19%, while 15 schools reported zero prevalence (Appendix A, Prevalence of *S. haematobium* by school).

#### 3.1.5. Factors Associated with *S. haematobium* among Schoolchildren

Table 3 presents the results of a logistic regression analysis between schistosomiasis infection status and potential risk factors. The adjusted analysis results indicate that the following variables were associated with schistosomiasis infection: being male; having a mother whose occupation is in livestock keeping; frequent contact with water; and the source of drinking water. Water contact behaviour may serve as a mediating factor between being male, the mother’s occupation, and the source of drinking water and schistosomiasis infection.

### 3.2. Adult Results

#### 3.2.1. Demographic Information of Adult Participants

A total of 1122 adult participants from 19 villages in the Itilima district participated in the present study. The median age of the study participants was 20 years (IQR: 17–30 years). Of these participants, 42.4% (476/1122) and 57.6% (646/1122) were male and female, respectively. Figure 3 shows the age and sex distributions of the study participants.

#### 3.2.2. Prevalence and Intensity of *S. haematobium* and Microhaematuria

Results from the urine filtration test technique were available for 1101 study participants. The overall prevalence of *S. haematobium* was 8.1% (95% CI, 6.6–9.8%). There was no significant difference in prevalence between sexes (χ^2^ = 0.77, *p* = 0.4, female 8.7% versus male 7.3%). Similarly, no significant differences in prevalence were observed among different age groups even though the youngest age group (15–25 years) had the highest prevalence at 8.5% (χ^2^ = 1.17, *p* = 0.8).

The overall arithmetic mean egg intensity, as determined by the urine filtration technique, was 1.91 eggs per 10 mL of urine (95% CI, 1.1–2.7). There was no significant difference observed between sexes (t = 0.42, *p* = 0.7). However, a significant difference was noted across age categories, with the 26- to 35-year age group exhibiting the highest mean egg intensity (mean = 5.3, F = 3.21, *p* = 0.02). According to the WHO categorization, 91% of the study participants with detectable eggs in their urine had a low intensity of infection, while 8.9% had a heavy intensity of infection.

#### 3.2.3. Prevalence of *S. haematobium* in Relation to the Village of Residence of Study Participants

Out of the 19 villages involved in the study, *S. haematobium* cases were found in 63.2% of them, with prevalence rates ranging from 1% to 57.9%. In the remaining seven villages, which constitute 36.8% of the total, no cases of *S. haematobium* were detected. The villages with the highest prevalence, compared to the others (χ^2^ = 291.53, *p* = 0.001), were Nangale (5.9%), Nkoma (11.1%), Luguru (12%), Mwamigagani (12.4%), Nhobora (18%), Budalabujiga (26.1%), and Sasago (57.9%). The prevalence in the remaining villages was less than 5%. No cases of *S. haematobium* were detected in the following villages: Laini, Dasina, Habiya, Kinang’weri, Mwamapalala, Mwaswale, and Ng’wabuki (Appendix A, Prevalence of Schistosoma haematobium by village).

#### 3.2.4. Prevalence and Infection Intensity of *S. mansoni*

The overall prevalence of *S. mansoni* was 0.45% (95% CI, 0.15–1.05%), and the overall geometrical mean egg intensity per gram of faeces (GMepg) was 41.3 Gmepg (95% CI, 6.3–268.7). Based on the formalin-ether concentration technique, *S. mansoni* eggs were detected in only one participant.

#### 3.2.5. Prevalence of STH Infection and Other Gastrointestinal Parasitic Infection

Based on both the Kato Katz technique and formalin-ether concentration technique, none of the study participants were detected to have any species of STH among both children and adults. Other parasitic infections detected using the formalin-ether concentration technique were *E. histolytica/E. dispar* (27.8%) and *Strongyloides* larvae (2%) among adults.

## 4. Discussion

Assessing the prevalence of *S. haematobium*, *S. mansoni,* and STHs (*T. trichiura*, *A. lumbricoides*, hookworms, and *Enterobius vermicularis*) is an essential prerequisite for the planning and implementation of cost-effective MDA. Here, we present the results of a baseline survey conducted in the Itilima district of north-western Tanzania prior to the implementation of an integrated intervention. This cross-sectional survey reveals that *S. haematobium* continues to pose a public health concern among school-aged children and the adult population in the study district. The results further indicate a geographical variation in the prevalence of *S. haematobium* between schools and villages within the same district. These findings suggest the potential existence of hot spots for *S. haematobium* transmission in certain districts. Infection with *S. haematobium* among schoolchildren was associated with the age group of 11–15 years, the mother’s occupation as a livestock keeper, the frequency of water contact per week, and the household’s source of drinking water. Conversely, the prevalence and intensity of *S. mansoni* and STHs were very low among schoolchildren and the adult population, with no adults detected as being infected with any of the STHs.

The overall mean school prevalence of *S. haematobium* observed in the present study was 10.1% among schoolchildren. This prevalence is slightly lower than the 12% previously recorded among schoolchildren in the same district over the past year by Mazigo et al. (submitted). The prevalence noted in this study was higher than the total prevalence of 8.9% reported on Pemba Island but lower than the 12% reported on Unguja Island among schoolchildren [21]. When compared to prevalence findings from other African countries, the overall prevalence of *S. haematobium* observed in this study was higher than the 8.7% reported in Nigeria [22] but lower than the 10.4% in Malawi [23], 60% in Zimbabwe [24], and 77.4% in Nigeria [25].

In the adult population, the overall prevalence of *S. haematobium* eggs was 8.1% and that of microhaematuria was 9.8%. The observed prevalence of microhaematuria was lower than the levels of 10.4% and 14.3% recorded among surveyed adults in Unguja and Pemba Islands [21]. However, the overall *S. haematobium* infection prevalence was 2.7% and 5.5% in Unguja and Pemba Islands [21]. The geographical variability in schistosomiasis prevalence in schoolchildren and adults can partly be explained by variations in ecological conditions, such as the proximity of infested water bodies and snail density in the areas where the study was conducted [12,26,27]. In addition, differences in the levels and frequency of water contact may account for the observed discrepancies [23].

In north-western Tanzania, the most recent mapping of schistosomiasis was conducted in 2004–2005, utilizing a combination of parasitological methods, a red urine questionnaire, and a geographical information system [12,28]. These surveys provided valuable epidemiological data on the distribution of *S. haematobium* in the region [12,28]. The prevalence of *S. haematobium* varied significantly among schools and villages. Eight schools had an *S. haematobium* prevalence rate exceeding 20%, and the surrounding communities also exhibited high prevalence rates, reaching up to 45%. Previous mapping identified Itilima as an area endemic for *S. haematobium*, with prevalence rates varying among schools [12,14]. Geographical variations in the prevalence of *S. haematobium* infection have been previously observed in Tanzania [21] and Malawi [23]. For example, in Unguja, the highest *S. haematobium* prevalence of 26.8% among schoolchildren was found in Shehia Upenja, and a prevalence of 20.0% was found among children from Shehia Kinyasini [21]. Among adults, the highest *S. haematobium* prevalence was found in Shehia Koani (26.5%) in Unguja and in Shehia Uwandani (23.4%) in Pemba [21]. These geographical variations in *S. haematobium* prevalence can be partially attributed to differences in social-ecological factors among villages. In north-western Tanzania, the transmission of *S. haematobium* is primarily influenced by two major factors: the presence of suitable water habitats containing the main intermediate hosts, *Bulinus nasutus* and *B. globosus* [29,30,31], and environmental contamination with human urine coupled with high human–water contact activities such as bathing, washing clothes, and agriculture [31]. The absence of snail habitats near schools or villages, the remote location of schools or villages from water bodies, and reduced water contact by schoolchildren and adults may explain the lack of detected cases of *S. haematobium* in some of the participating schools or villages. This is consistent with findings reported on Unguja Island [26]. These findings have implications for the planning and implementation of MDA. Not all schools located in these districts qualify for the annual MDA as it is currently practiced. Implementing MDA once a year or adopting a treatment strategy that takes into account the transmission season of *S. haematobium* infection may help to reduce costs.

In the present study, several factors were found to be associated with *S. haematobium* infection. These included the age of the schoolchildren, the occupation of their mothers, the frequency of water contact, and the primary source of drinking water for the household. Some of these factors have been previously reported as being associated with *S. haematobium* infection in Malawi [23] and Zanzibar [21]. The key observation from this study is that most of these factors can be revised or modified if a targeted control intervention is implemented. In the questionnaire survey conducted among schoolchildren, the majority reported that the main sources of drinking water for their households and schools were open and closed wells and rivers, with only 20% reporting the availability of piped water at their homes. This underscores the need for water, sanitation, and hygiene improvement interventions to be integrated into MDA programs. Improvements in water and sanitation infrastructure, along with the use of clean water and latrines by the population, are essential components of an integrated control intervention. This is particularly true if the goal is to eliminate *S. haematobium* infection in the district.

The prevalence of *S. mansoni* in both schoolchildren and the adult population was notably low (below 1%). The existing literature presents contrasting results regarding the relationship between the proximity to the Lake Victoria basin and the prevalence of schistosomiasis. It has been reported that those residing in communities closest to the lake have the highest prevalence of *S. mansoni* infection, while those living relatively far from the lake shoreline have the highest prevalence of *S. haematobium* infection [12,32]. Conversely, the prevalence of *S. haematobium* tends to increase with distance from the lake shoreline, while that of *S. mansoni* decreases. This observation aligns with the findings of Mugono et al. [32] in north-western Tanzania and Handzel et al. in western Kenya [33]. This pattern can be partially explained by the preferences of the intermediate hosts involved in the transmission of the two parasites. The hosts involved in the transmission of *S. mansoni* tend to prefer large water bodies, while those transmitting *S. haematobium* favour smaller, inland, and seasonal water bodies [30].

The overall prevalence of STHs was extremely low (below 0.5%) among schoolchildren, with no infection detected in any of the adults. This finding aligns with the results of previous studies by Lwambo et al. [34] and Clements et al. [35], which suggest that STH infections outside the Lake Victoria basin are either very low or negligible. In this region, STH infections are predominantly found in the highland areas located on the western highlands of Lake Victoria [36].

The geographical prevalence of *S. haematobium* infection in our study has several implications for MDA in terms of planning, implementation, and evaluation, with the ultimate goal of reducing the schistosomiasis map in Tanzania. The WHO has established criteria for schistosomiasis prevalence and morbidity control [37], setting an ambitious goal to eliminate schistosomiasis as a public health problem (defined as a <1% proportion of heavy-intensity infections) in all endemic countries by 2030 [38]. Therefore, identifying at-risk populations and mapping hotspot areas for targeted intervention is one of the WHO’s objectives to achieve the 2030 goals. According to this criterion, morbidity control can be achieved if the prevalence of heavy infection intensity with any schistosome species is reduced to <5% [37], and for MDA, it is achieved wherever the schistosomiasis prevalence exceeds 10% [37]. Following this recommendation, some school-aged children, especially those attending schools with zero prevalence, will not require annual MDA. Conversely, schools recording a prevalence of <10% may not need annual MDA. However, schools classified as moderate-risk (prevalence ≥ 10%–< 50%) [39] would require treatment once every two years [37]. To significantly reduce prevalence and interrupt transmission in areas classified as moderate-risk, it will be crucial to intensify intervention measures in these areas in the future. Alternatively, to interrupt transmission in the future, a well-focused treatment strategy needs to be designed, focusing solely on schools classified as moderate-risk. This proposed strategy should concentrate on designing an integrated intervention.

## 5. Conclusions

The current study provides data on the geographical distribution of *S. haematobium*, *S. mansoni*, and STHs in the Itilima district, north-western Tanzania. *S. haematobium* remains a public health concern, with its prevalence varying from one school or village to another. The prevalence of the other infections investigated is below 1%. This study observed a geographical variation in the prevalence of *S. haematobium* between schools and villages, identifying hot-spot locations where prevalence exceeded 20%. *S. haematobium* infection was associated with certain factors, including children’s age groups (11–15 years), maternal occupation, livestock keeping, frequency of water contact, and the household’s source of drinking water. These factors indicate access and exposure to risk environments. To control and eliminate *S. haematobium* infection in the study setting, complementary measures are recommended. These include health education, improving safe water supply, and providing adequate sanitary facilities.

## Figures and Tables

**Figure 1 life-13-02333-f001:**
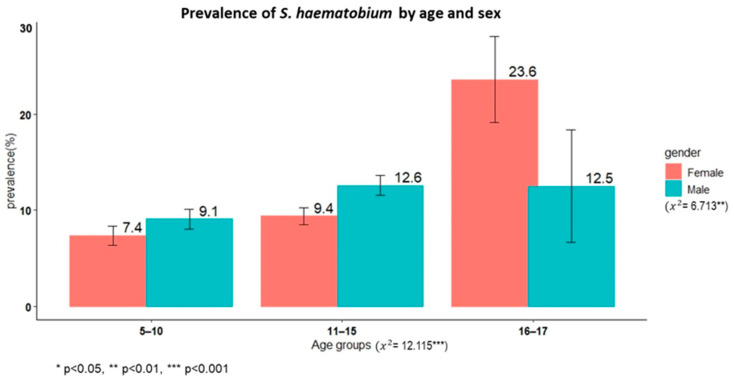
Prevalence of *S. haematobium* by age group among primary schoolchildren in Itilima district, north-western Tanzania, 2020.

**Figure 2 life-13-02333-f002:**
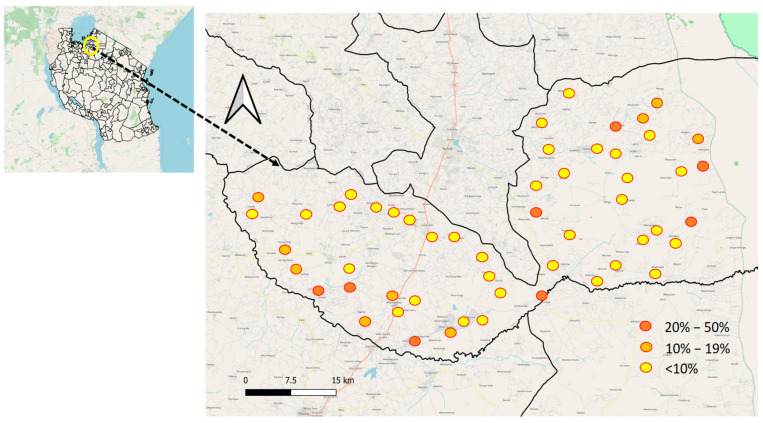
Geographical distribution of *S. haematobium* prevalence at the school level in Itilima district, 2020.

**Figure 3 life-13-02333-f003:**
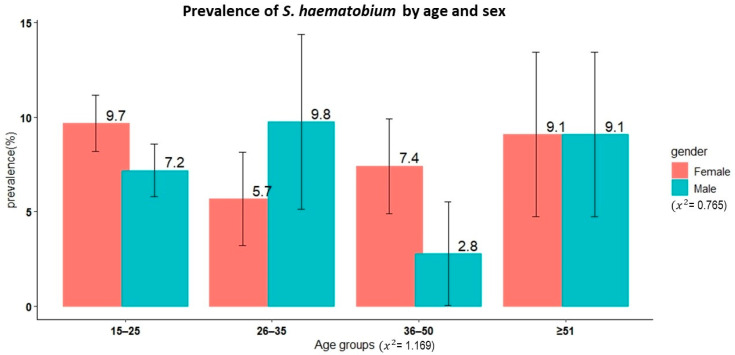
Age and sex distribution of the adult study participants from 19 villages in Itilima district, north-western Tanzania, 2020.

**Table 1 life-13-02333-t001:** Inclusion/exclusion criteria.

Participants’ Attributes	Reason
Inclusion criteria: children will be included if the following holds:	
Aged 3–17 years of age	The age group with high prevalence and infection intensity For the primary school to have pupils aged over 15 years old depends on several reasons 1. Late enrolment to school due to family matters like low income to fulfil educational materials to support the child, child health problems, or other matters like assisting with domestic chores at a young age. 2. The child repeating the same grade for the following year due to low academic performance.
Permanent residence of the selected areas/schools	At risk of infection
No history of anthelminthic treatment in the past 6 months	To avoid effects of previous treatment
They have provided single urine and stool	For parasitological diagnosis
Provide signed parental consent for the children to participate in the study	Allowing children to be accepted in the study and adhering to the Helsinki Declaration; participation in the study is on voluntary basis
Adult individuals will be included in the study if the following holds:	
Aged 15–60 years	Depending on the force of infection, they remain susceptible to infection
Provide written informed consent	To ensure voluntary participation in the study
Submit urine and stool samples	For parasitological screening
No history of using anthelminthic in the past 6 months	To avoid effects of previous treatment
Exclusion criteria: children and adult will be excluded from the study if the following holds:	
Reported to be allergic to the drug used in the study (praziquantel drug)	To avoid health effects related to the allergic response due to treatment
They have received anthelminthic treatment outside of the study during the follow-up period	To avoid measuring the effects of other treatment rounds
Parents/guardians refuse to give informed consent	Respect parental decisions/participation is voluntary
Failure to provide stool and urine samples	To allow baseline collection of parasitological data

**Table 2 life-13-02333-t002:** General characteristics of participants (schoolchildren).

Variables			Variables		
Child Age		11.2 (2.6)	Toilet existence (HH)		94.5% (3016/3191)
Child Gender (female)		50.0% (1889/3779)	Regular use of toilet (HH and school)		89.9% (2870/3191)
Education level (caregivers)	Illiterate (no education)	8.4% (267/3191)	Latrine type (HH)	Pit latrine	93.9% (2831/3016)
	Not completed primary	19.7% (630/3191)		Flush toilet	6.1% (185/3016)
	Completed primary	69.4% (2216/3191)	Source of water (home)	Unprotected spring	39.7% (1266/3191)
	Completed secondary	2.4% (78/3191)		Hand pump wells	26.3% (839/3191)
				Portable piped	18.8% (599/3191)
Education level (household head)	Illiterate (no education)	5.8% (186/3191)		River	7.5% (238/3191)
	Not completed primary	16.0% (509/3191)		Spring	7.3% (234/3191)
	Completed primary	66.9% (2134/3191)	Handwashing after defecation		74.3% (2372/3191)
	Completed secondary	11.0% (362/3191)	Existence of toilet (school)		89.9% (2870/3191)
Father’s occupation	Farmer/peasants	66.8% (2133/3191)	How often do you contact water	None	41.3% (1319/3191)
	Livestock keeper	14.4% (460/3191)		Once a week	4.2% (133/3191)
	Small business scale	10.5% (334/3191)		Two times a week	5.4% (172/3191)
	Employed	4.2% (133/3191)		Three times a week	6.6% (210/3191)
	Household mother/father	1.8% (56/3191)		More than 3 times per week	42.5% (1357/3191)
Place of defecation (both)	Toilets	94.5% (3017/3191)	Having dewormer at school or community	at school (yes)	76.2% (2432/3191)
	Bush/rivers	5.5% (174/3191)		at community (yes)	20.1% (642/3191)

Regular use of toilet = daily use of a toilet for urination and defecation.

**Table 3 life-13-02333-t003:** Logistic regression tables for factors associated with *Schistosoma haematobium* among schoolchildren at Itilima district council, north-western Tanzania.

Variable	OR	95% CI	*p*-Value	aOR	95% CI	*p*-Value
Sex of the student						
Female	REF					
Male	1.03	1.01–1.05	0.007	1.02	1.01–1.04	0.03
Age group in years						
<5	REF					
5–9	1.08	0.60–1.96	0.80	1.07	0.59–1.93	0.82
10–14	1.12	0.62–2.03	0.72	1.10	0.61–2.00	0.75
≥15	1.13	0.62–1.96	0.68	1.12	0.61–2.02	0.72
Mothers’ education level						
Illiterate (never went to school)	REF					
Illiterate	0.95	0.91–0.98	0.004	0.96	0.92–0.99	0.02
Fathers’ occupation						
Other	REF					
Farmer	0.97	0.95–0.99	0.02	0.97	0.95–0.99	0.02
Open defecation (self-report)						
Toilets	REF					
Bushes/rivers	1.02	0.98–1.07	0.34	1.02	0.97–1.07	0.44
Toilet at household						
No	REF					
Yes	0.96	0.92–1.01	0.11	0.99	0.93–1.05	0.80
Toilet at school						
No	REF					
Yes	0.96	0.93–0.99	0.02	0.97	0.93–1.01	0.13
Type of toilet (at house)						
No toilet	REF					
Pit latrine	0.96	0.78–1.08	0.59	0.95	0.82–1.11	0.54
Flush toilet	0.92	0.82–1.12	0.31	0.93	0.79–1.09	0.36
Water contact behaviour						
None	REF					
Once a week	1.04	0.99–1.10	0.11	1.04	0.98–1.09	0.19
Two times a week	1.06	1.01–1.11	0.01	1.05	1.01–1.11	0.02
Three times a week	1.09	1.05–1.14	<0.001	1.08	1.04–1.13	<0.001
More than three times a week	1.06	1.04–1.09	<0.001	1.06	1.03–1.08	<0.001

## Data Availability

Data are available upon request.

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
