# Peer review of "Prevalence of Schistosomiasis and Soil-Transmitted Helminthiasis and Their Risk Factors: A Cross-Sectional Study in Itilima District, North-Western Tanzania"

_life, 2023, doi:10.3390/life13122333_

Round 1

Reviewer 1 Report

Comments and Suggestions for Authors

This is an interesting manuscript and I have the following comments the authors may like to consider.

Abstract

What does ‘and associated factors at Itilima district, north-western Tanzania’ mean? What associated factors?  I am unsure what the authors mean here.  Do you mean what factors are associated with infection?

Are you looking at every soil associated helminth in this study or specific ones? If so can you gibe the species names?

S. haematobium needs to be in italics and a space is required between the two names.

What do the authors mean by ‘In total, 3,779 schoolchildren 27 had complete results from urine testing’? You only had results from 27 school children?

Do you have any p values for your correlations? They should be quoted so the reader knows these correlations are statistically significant.   

How do the authors reconcile the statement that ‘The prevalence of S. mansoni and soil-transmitted helminths was low among both children and adults’ with the statement at the end that this study helps with ‘implementation of mass drug administration’? If the incidence is very low then I would have thought a mass drug program was not required?

Introduction.  Well written and clear

Materials and methods

Ethical approval – I would put this section near the beginning of this section rather than at the end.

I see the authors used parametric statistics to analyse some of their data – ‘The mean egg counts for S. mansoni between sex and age groups were compared using either the t-test (for two groups) or ANOVA (for more than two groups’.  I am a bit surprised as I would not expect this type of data to be normally distributed and this is one of the criteria for analysing data using parametric tests.  The authors should state how they showed their data was normally distributed and if they did not consider this factor then the data needs re-analysing using non-parametric statistical tests. It may give the same conclusions – but data must be analysed correctly.

Results

Table 2

There are subheadings on the right of the table that are not clear. What does ‘Regular use of toilet’ mean.  You may be better to split this table and make things clearer to the reader.

Fig. 1 Remove the numbers off the columns.  I presume there are no significant differences between groups as none are shown.  Are the older children more likely to be infected? You need a final number on the y axis – so not have it going into infinity. I presume it should have 30.

Table 3.  Species name should be in italics in the legend. I cannot see how you have two p values on a row.  If you are comparing male and female then there is only two treatments?  Some of these p values may be related to Fig 1.  If this is the case then the p values should go on the data.  What is the REF value in this table? This should be explained in the legend.  I am not sure what you are comparing -  e.g. mother’s literacy is being compared to what?

Fig. 3 Are there any significant differences between groups? Is this shown in Table 4 – in which case there are no significant differences between the groups and Table 4 is redundant.  You could just quote the Chi Square values in the figure legend if you wanted to give them.  Take the numbers off the columns.

You do not give any species for STH – is that because you could not find out? You name some in the introduction.  Ascaris eggs and hookworms eggs are distinctive. So that seems surprising.

Comments on the Quality of English Language

This is a well written manuscript with the odd bit that is not quite clear

Author Response

We have successfully responded to all the comments

Reviewer 2 Report

Comments and Suggestions for Authors

Attached

Comments on the Quality of English Language

Author Response

The response to reviewer comments are attached
